# Utilizing Envelope Analysis of a Nasal Pressure Signal for Sleep Apnea Severity Estimation

**DOI:** 10.3390/diagnostics13101776

**Published:** 2023-05-17

**Authors:** Mikke Varis, Tuomas Karhu, Timo Leppänen, Sami Nikkonen

**Affiliations:** 1Department of Technical Physics, University of Eastern Finland, Canthia, P.O. Box 1627, FI-70211 Kuopio, Finland; tuomas.karhu@uef.fi (T.K.); timo.leppanen@uef.fi (T.L.); sami.nikkonen@uef.fi (S.N.); 2Diagnostic Imaging Center, Kuopio University Hospital, FI-70210 Kuopio, Finland; 3School of Information Technology and Electrical Engineering, The University of Queensland, Brisbane, QLD 4072, Australia

**Keywords:** sleep apnea, envelope analysis, objective analysis, nasal pressure, severity estimation, respiratory event

## Abstract

Obstructive sleep apnea (OSA) severity assessment is based on manually scored respiratory events and their arbitrary definitions. Thus, we present an alternative method to objectively evaluate OSA severity independently of the manual scorings and scoring rules. A retrospective envelope analysis was conducted on 847 suspected OSA patients. Four parameters were calculated from the difference between the nasal pressure signal’s upper and lower envelopes: average (AV), median (MD), standard deviation (SD), and coefficient of variation (CoV). We computed the parameters from the entirety of the recorded signals to perform binary classifications of patients using three different apnea–hypopnea index (AHI) thresholds (5-15-30). Additionally, the calculations were undertaken in 30-second epochs to estimate the ability of the parameters to detect manually scored respiratory events. Classification performances were assessed with areas under the curves (AUCs). As a result, the SD (AUCs ≥ 0.86) and CoV (AUCs ≥ 0.82) were the best classifiers for all AHI thresholds. Furthermore, non-OSA and severe OSA patients were separated well with SD (AUC = 0.97) and CoV (AUC = 0.95). Respiratory events within the epochs were identified moderately with MD (AUC = 0.76) and CoV (AUC = 0.82). In conclusion, envelope analysis is a promising alternative method by which to assess OSA severity without relying on manual scoring or the scoring rules of respiratory events.

## 1. Introduction

Obstructive sleep apnea (OSA), characterized by complete (apnea) and partial (hypopnea) breathing cessations, is a prevalent sleep disorder affecting nearly a billion adults globally [1]. Currently, the diagnosis of OSA and the assessment of its severity are mainly based on the presence of daytime sleepiness and the apnea–hypopnea index (AHI) [2,3]. Although the AHI is the most commonly used diagnostic parameter, it has several shortcomings [4]. For example, the scoring of respiratory events is performed manually based on arbitrary and somewhat vague visual scoring rules [5]. Hence, the determination of AHI and the evaluation of OSA severity is time-consuming and prone to human error.

To overcome some of the limitations of the AHI, more sophisticated parameters have been developed for a better assessment of OSA severity [6,7,8,9,10,11]. However, most of these parameters are still based on manual scorings, and thus, are limited by the arbitrary scoring rules and the subjectivity of the scorers. Additionally, to reduce the need for manual scoring, several different machine learning methods have been developed to automatically estimate the severity of OSA [12,13,14]. Yet, the performance of these methods is still ultimately reliant on the manual scorings which are used to train the models.

One solution to decrease the reliance on AHI and manual scoring rules could be the evaluation of OSA severity with the help of envelope analysis of an airflow signal. The amplitude of an airflow signal decreases during a respiratory event [5], leading to a smaller difference between its upper and lower envelopes. Therefore, as the nasal pressure signal has been shown to be sensitive enough to detect even mild disturbances in respiration [15], features of its envelopes could provide additional information on the severity of OSA. Consequently, algorithms based on the nasal pressure signal’s envelopes have been used in automatic and accurate respiratory event detection [16,17]. Furthermore, Diaz et al. [18] have previously introduced a parameter computed from the upper envelope of the nasal pressure signal which was able to quantify disordered breathing during sleep. However, parameters calculated directly from the airflow signal envelopes have not been widely utilized in OSA severity assessment.

The aim of the present study was to provide an alternative method for OSA severity assessment that avoids the known issues in AHI-based estimation. The presented method is based on the parameters calculated from the envelopes of the nasal pressure signal that are independent of the manual scorings or scoring rules of respiratory events. We hypothesize that smaller values and greater variation in the difference between the nasal pressure signal upper and lower envelopes could imply more disturbed breathing and more severe OSA. The performance of the presented envelope method was evaluated in a large clinical OSA population.

## 2. Methods

### 2.1. Study Population

A total of 887 suspected OSA patients had undergone type 1 polysomnography (PSG) in the Princess Alexandra Hospital (Brisbane, Australia) during 2015–2017. The PSGs included recordings of abdominal and thorax respiratory efforts, electroencephalography (6 channels), electrooculogram (2 channels), electrocardiogram, chin and leg electromyography, airflow with nasal pressure and thermocouple, photoplethysmography, body position, snoring, sound, oxygen saturation, and transcutaneous carbon dioxide. All the signals were recorded and analyzed with the Compumedics Grael acquisition system and Compumedics ProFusion 4.0 software (Compumedics, Abbotsford, Australia). Approval for the data collection and processing was provided by the institutional human research ethics committee of the Princess Alexandra Hospital (HREC/16/QPAH/021 and LNR/2019/QMS/54313). All PSGs were manually scored by the sleep technicians in compliance with the American Academy of Sleep Medicine (AASM) guidelines using a 3% desaturation or an arousal rule for hypopnea scoring [19]. From the full PSGs, only the nasal pressure signals were utilized in the envelope calculations and analysis. Based on the visual inspection, patients with failed nasal pressure recording were excluded from this study (*N* = 40). The recording was considered to have failed if it consisted mostly of noise or if the nasal cannula was disconnected for most of the recording duration. Thus, the final study population comprised 847 patients. The demographic and PSG information of the study population is presented in Table 1.

### 2.2. Signal Preprocessing

The nasal pressure signals were recorded with a 128 Hz sampling frequency. All the segments where the nasal cannula was detached for more than a second were removed from the signals. After artifact removal, signals were filtered with a 4th order Butterworth low-pass filter with a 3 Hz cutoff frequency. Furthermore, as the signal amplitudes may vary significantly between the patients and within the night due to changes in body positions and sleep stages [19], signals were Z-score normalized using a 5-min sliding window with a window step size of one data point. These normalized signals will be referred to as preprocessed signals for the remainder of this paper. All the signal preprocessing steps and analyses were conducted using Matlab (version R2022a, MathWorks, Natick, MA, USA).

### 2.3. Envelope Algorithm

First, to locate the signal’s local extreme points, the preprocessed signal was smoothed with a 0.5-s moving average filter. However, the detected locations of extreme points did not exactly match the locations of the true local maximum and minimum values in the preprocessed signal. Thus, the locations were corrected by searching the true maximum and minimum values from the preprocessed signal within a 1-s window placed around the detected extreme points. Next, the upper envelope was formed by interpolating between the maximum points, while the lower envelope was formed by interpolating between the minimum points using a piecewise cubic hermite interpolating polynomial [20]. Finally, a difference envelope was formed by subtracting the lower envelope from the upper envelope. In addition, to prevent negative difference envelope values caused by intersecting upper and lower envelopes, difference envelope values during such periods were set to zero. In total, 0.064 % of all the difference envelope values were adjusted to zero. An illustration of the envelopes and the difference envelope is presented in Figure 1.

### 2.4. Data Analysis

We calculated the average (AV), median (MD), standard deviation (SD), and the ratio of SD to AV, i.e., coefficient of variation (CoV) from the difference envelope. The parameters were computed separately from the full-night envelopes and envelopes truncated in non-overlapping 30-s epochs.

We determined receiver operating characteristic (ROC) curves for the parameters calculated from the full-night difference envelope to assess their ability to perform binary classification of patients into two groups with three different AHI thresholds (AHI = 5, 15, and 30). Individual ROC curves were computed separately with each AHI threshold, resulting in three distinct ROC curves per parameter. Moreover, an additional ROC curve was determined to evaluate how well parameters can distinguish non-OSA patients from severe OSA patients. For all cases, corresponding areas under the curves (AUC) were calculated. Lastly, the correlations between the full-night difference envelope parameters and AHI were investigated.

ROC curves and corresponding AUCs were also determined for difference envelope parameters calculated within 30-s epochs to estimate whether or not an epoch contains a respiratory event based on the parameter values. An epoch was considered to contain a respiratory event if it included at least one data point of manually scored apnea or hypopnea. Furthermore, the percentage of epochs containing an event with a specific range of parameter values was calculated. Moreover, the distributions of the parameter values in the 30-s epochs were determined for each OSA severity group. Finally, the effect of respiratory event duration on the parameter values was visualized.

## 3. Results

### 3.1. Full-Night Envelope Analyses

The AUCs for all parameters increased with increasing AHI thresholds between the groups (Figure 2). Among the four parameters, SD displayed the best overall differentiation ability with AUCs ≥ 0.86, whereas AV performed the worst with AUCs ≤ 0.77 for all three AHI thresholds. Moreover, patients with severe OSA were distinguished extremely well from non-OSA patients by utilizing the SD (AUC = 0.97) and CoV (AUC = 0.95). However, the correlations between the difference envelope parameters and the AHI were only moderate (Figure 3).

### 3.2. Epoch-by-Epoch Analyses

An epoch-by-epoch example of difference envelope parameter values from one patient is presented in Figure 4. Parameter values varied substantially during the recording depending on whether a respiratory event occurred in an epoch or not. Based on the ROC analyses, parameters calculated in the 30-s epochs were able to moderately separate epochs containing respiratory events from those without events (Figure 5). In addition, with increasing AV and MD values a smaller percentage of epochs included events (Figure 6). An opposite trend was observed with SD and CoV, as the percentage of epochs containing events increased with increasing parameter values.

The distributions of epoch-wise calculated difference envelope parameter values are shown in Figure 7. An evident difference was observed in the AV and MD values in which the severe OSA group had a noticeable peak close to zero (Figure 7a,b). Furthermore, the duration of a respiratory event within a 30-s epoch had a major impact on the parameter values (Figure 8). The median values decreased for AV and MD (Figure 8a,b) and increased for SD and CoV (Figure 8c,d) with longer event duration. However, an exception was observed in the SD in which epochs without events and epochs with 25–30 s events had similar values.

## 4. Discussion

In this study, we introduced an alternative method based on the parameters calculated from the envelopes of the nasal pressure signal to objectively assess the severity of OSA. We found that the SD (AUCs ≥ 0.86) and CoV (AUCs ≥ 0.82) performed best in the classification of patients with all three AHI thresholds. Furthermore, the manually scored respiratory events within the 30-s epochs were detected moderately using MD (AUC = 0.76) and CoV (AUC = 0.82). Overall, the results are in line with our hypothesis, as the mean of the difference envelopes decreased, and the variation increased with increasing OSA severity. 

In the full-night ROC analysis, the SD and CoV performed better for all AHI thresholds compared with the AV and MD (Figure 2). For example, using the AHI threshold of 5, the performances of the AV (AUC = 0.64) and MD (AUC = 0.70) when classifying patients into the correct group were only moderate, whereas much better performances were obtained with SD (AUC = 0.86) and CoV (AUC = 0.82). Furthermore, with the SD and CoV, severe OSA patients were distinguished extremely well from non-OSA patients with AUCs of 0.97 and 0.95, respectively. Thus, based on these results, the full-night SD and CoV are the most promising parameters among the four investigated variables for estimating the severity of OSA. However, as OSA is a vastly heterogeneous disease, using a single parameter to assess its severity is most likely insufficient [21]. Thus, a combination of these envelope parameters, or potentially some more advanced envelope parameters, with other relevant OSA severity metrics, could enable a more comprehensive evaluation of its severity.

The effect of respiratory events on epoch-wise calculated envelope parameter values is clear. The percentage of epochs containing respiratory events progressively decreased with increasing AV and MD values and increased with increasing SD and CoV values (Figure 6). In addition, the patients with more severe OSA had more epochs with smaller AV and MD values (Figure 7a,b). This finding is consistent with our hypothesis that more disturbed breathing leads to smaller AV and MD values within the epochs. Furthermore, the duration of the respiratory events had a major effect on all of the parameter values (Figure 8). However, as the duration of respiratory events can range from 10 s up to several minutes [22], the 30-s epochs may be too short, especially for SD, to efficiently capture changes in the difference envelope. This can be deduced from the observation that the epochs without respiratory events and epochs with 25–30 s events had similar SD values (Figure 8c). However, this is reasonable because the difference envelope values should stay relatively stable during regular cyclical breathing and during a long respiratory event that can last the whole 30-s epoch. 

Based on the 30-s epoch ROC analyses, the existence of respiratory events was only moderately identified with the epoch-wise computed parameter values (Figure 5). However, an epoch was considered to contain a respiratory event even when it included only one data point of a manually scored event. Therefore, an epoch containing only a few data points of a manually scored event likely does not cause any significant changes in the parameter values, and thus, correct classification of such epochs is challenging. Moreover, since the analyses were performed based on the manually scored respiratory events and their arbitrary rules [5], abnormal breathing cessations and patterns which do not satisfy the respiratory event criteria may still be classified as events by envelope parameters. Such instances include, for example, breathing cessations lasting less than 10 s or unscored events. However, the utilization of envelope analysis enables the detection of such abnormal events rather than just classifying events into apneas and hypopneas. In addition, envelope analysis takes into account the duration of respiratory events, unlike the AHI. Thus, although we compared the envelope parameters with the AHI, a perfect correlation was not the desired outcome as there are several well-known weaknesses in the AHI-based OSA severity estimation [4,23,24,25]. For these reasons, envelope analysis has the potential to enhance the assessment of OSA severity by revealing its true severity in more detail.

The present study has some limitations. We restricted our analysis to the nasal pressure signal, although breathing effort was also measured with a thermocouple and abdominal and thorax belts. The nasal pressure signal was selected as it is sensitive enough to detect even mild disturbances in respiration [15]. However, the presented envelope method could also be applied to other respiratory signals requiring no or only minimal changes in preprocessing and artifact detection. For example, applying the envelope analysis simultaneously to the respiratory effort signals may enable the separation of obstructive and central breathing disturbances and allow a more comprehensive evaluation of the nature of sleep apnea. Furthermore, the patients with failed nasal pressure recording (*N* = 40) were excluded from the study based on visual inspection. We opted to use visual inspection rather than some objective criteria, as automatic signal quality and artifact detection was not the focus of this study. In addition, as we only removed the signal segments where the nasal cannula had been disconnected, the analyzed signals may have contained several other artifact types which may affect the results. However, this can also be seen as a strength of the present study as the quite straightforward envelope method performed well without heavy signal processing. This need for minimal artifact removal should allow the presented method to be more easily applied to other datasets without major modifications. In the epoch-by-epoch analyses, we used non-overlapping epochs with a duration of 30 s. Thus, depending on the parameter, the duration of the epoch may not be optimal, e.g., too short for SD or too long to optimally detect short events. However, the parameter calculation would be very simple to undertake for epochs of any arbitrary length. Finally, we evaluated the performance of the envelope analysis by comparing parameter values with the AHI and manually scored respiratory events, despite their known limitations. However, a comparison with the current OSA standards is appropriate as the aim of the present study was to introduce a different approach to the evaluation of the severity of nocturnal breathing disturbances alongside the AHI. Thus, the aim was not to show superiority over AHI or to suggest replacing AHI-based OSA severity assessment with the presented envelope methods. 

In conclusion, envelope analysis of the nasal pressure signal performed relatively well in evaluating OSA severity. The introduced method is completely objective as it does not rely on the manual scoring of the respiratory events or their arbitrary scoring rules. Hence, given the widely known issues in OSA severity assessment, the envelope analysis of the airflow signal is a promising method that could enhance the assessment of nocturnal breathing disturbances.

## Figures and Tables

**Figure 1 diagnostics-13-01776-f001:**
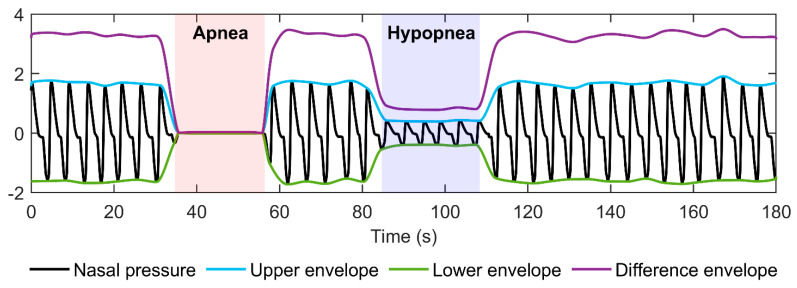
An illustration of nasal pressure signal envelopes and the difference envelope.

**Figure 2 diagnostics-13-01776-f002:**
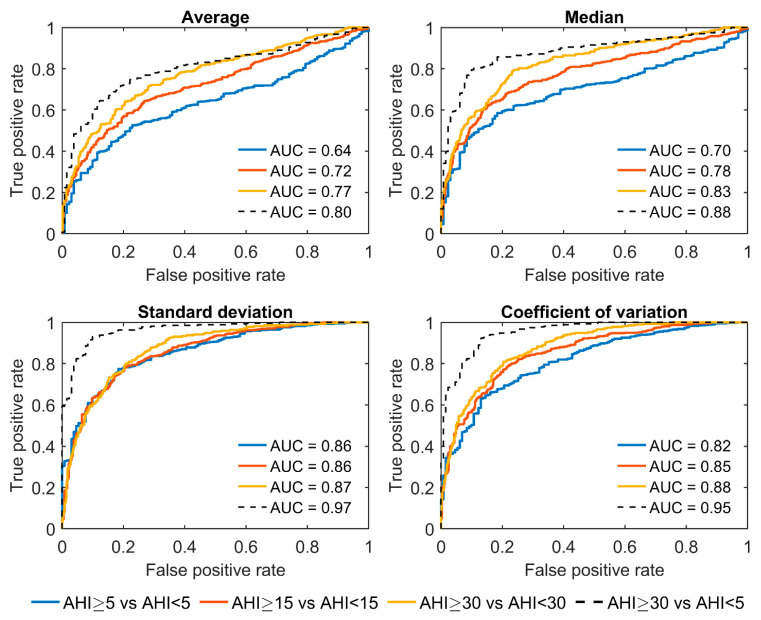
Receiver operating characteristic curves and corresponding areas under the curves (AUC) for binary classification of patients into two groups using three different AHI thresholds (AHI = 5, 15, and 30). The classification was also assessed only between severe (AHI ≥ 30) and non-OSA (AHI < 5) patients. Parameters were calculated from the full-night difference envelope.

**Figure 3 diagnostics-13-01776-f003:**
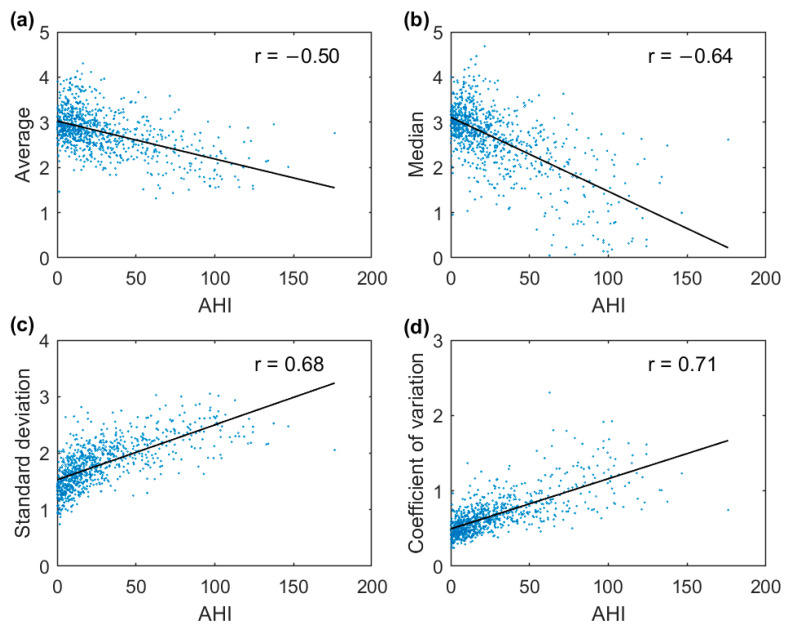
Correlations between the apnea–hypopnea index (AHI) and the average (**a**), median (**b**), standard deviation (**c**), and coefficient of variation (**d**) of the full-night difference envelope.

**Figure 4 diagnostics-13-01776-f004:**
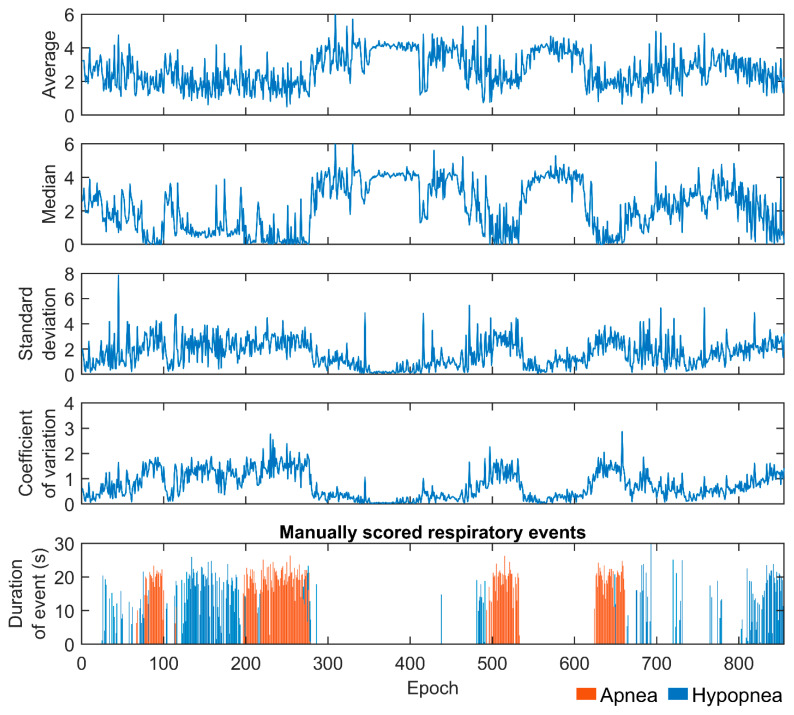
Epoch-by-epoch illustration of difference envelope parameter values and the manually scored respiratory events. The illustration represents a single patient whose manually determined apnea–hypopnea index was 58.6 events/h.

**Figure 5 diagnostics-13-01776-f005:**
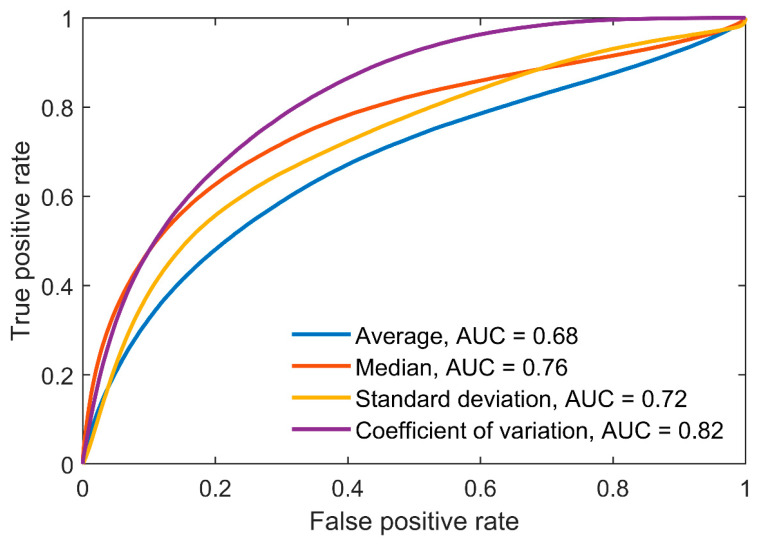
Receiver operating characteristic (ROC) curves and corresponding areas under the curves (AUC) for difference envelope parameter values calculated from 30-s epochs. ROC curves illustrate the ability of the parameters to separate epochs with and without respiratory events. An epoch was considered to include a respiratory event if it contained at least one data point of a manually scored event. Apneas and hypopneas were not differentiated.

**Figure 6 diagnostics-13-01776-f006:**
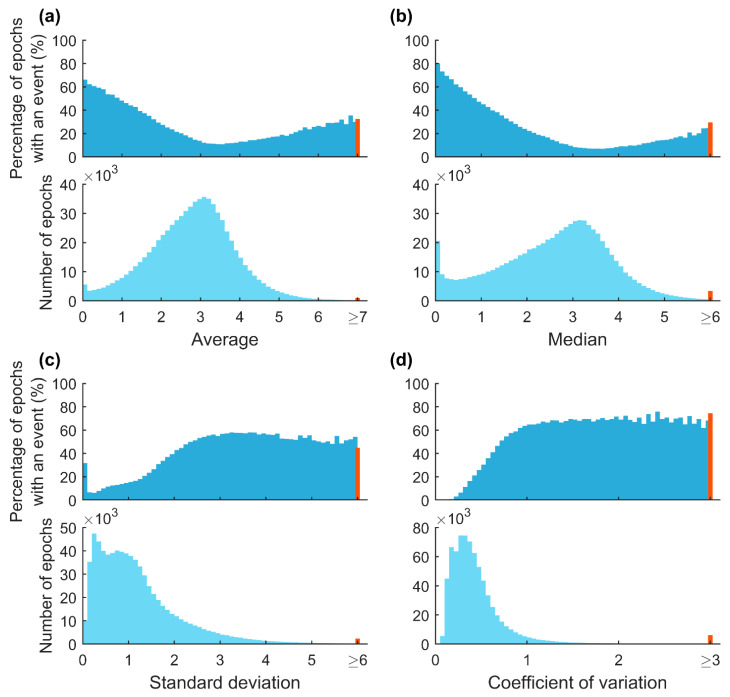
Relationship between percentages of the 30-s epochs containing manually scored respiratory events and parameter values. The red bins represent all of the extreme values of each parameter. The maximum parameter values were 11.8 for average (**a**), 13.2 for median (**b**), 14.5 for standard deviation (**c**), and 33.6 for coefficient of variation (**d**).

**Figure 7 diagnostics-13-01776-f007:**
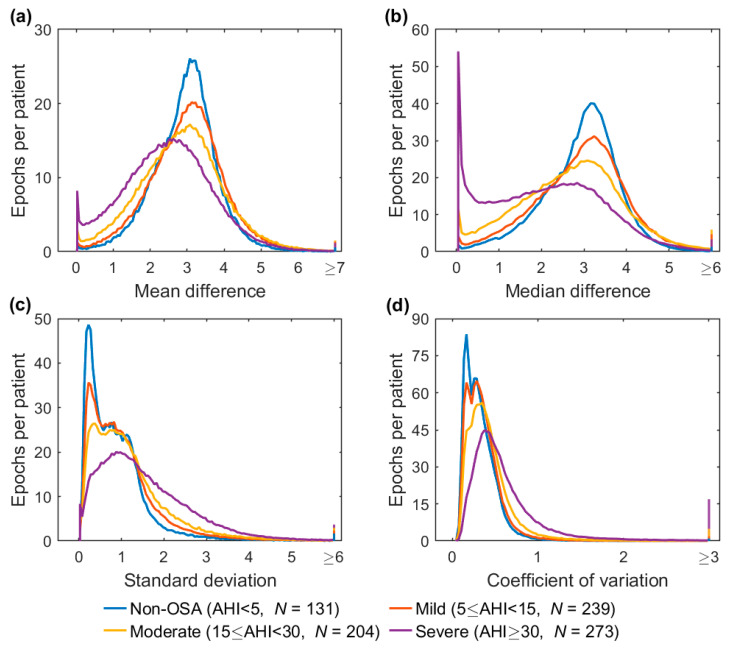
Distributions of the difference envelope parameter values calculated from 30-s epochs in different obstructive sleep apnea (OSA) severity groups. The distributions were normalized by the number of patients (*N*) in their respective severity groups. The last bins represent all of the extreme values of each parameter in the OSA severity group. The maximum parameter values were 11.8 for average (**a**), 13.2 for median (**b**), 14.5 for standard deviation (**c**), and 33.6 for coefficient of variation (**d**). Note the different scales in the *y*-axes.

**Figure 8 diagnostics-13-01776-f008:**
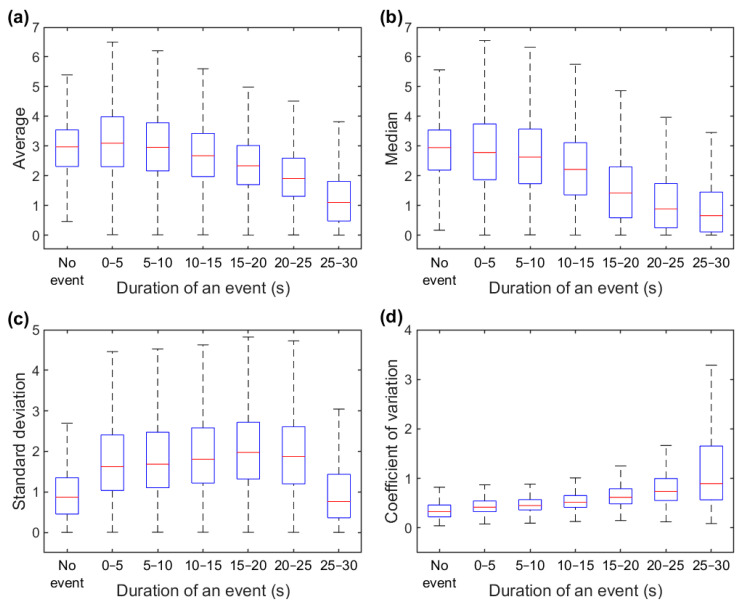
Relationship between respiratory event duration and average (**a**), median (**b**), standard deviation (**c**), and coefficient of variation (**d**) values calculated from difference envelope within 30-s epochs. The red line displays the median value, the edges of the boxes show the interquartile range (IQR), and the whisker the range of the data without outliers. Outliers were defined as values over 1.5 times IQR from the upper quartile and values below 1.5 times IQR from the lower quartile. Outliers are not presented.

**Table 1 diagnostics-13-01776-t001:** The demographic and polysomnographic information of the study population.

	Whole Population	Non-OSA (AHI < 5)	Mild OSA (5 ≤ AHI < 15)	Moderate OSA (15 ≤ AHI < 30)	Severe OSA (AHI ≥ 30)
Patients (*N*, (male%))	847 (53.8)	131 (32.8)	239 (45.2)	204 (55.9)	273 (70.0)
Age (years)	55.8 (44.8–65.7)	44.8 (31.4–58.3)	54.4 (44.8–64.1)	56.8 (48.2–66.5)	59.0 (48.0–68.5)
BMI (kg/m^2^)	34.0 (29.0–40.2)	30.0 (25.0–35.3)	33.6 (28.3–38.9)	33.7 (30.3–39.9)	36.2 (31.8–43.1)
AHI (events/h)	18.0 (8.2–38.4)	2.5 (1.3–3.6)	9.8 (7.2–12.2)	21.3 (17.9–25.0)	56.0 (39.4–78.0)
Duration of analyzed period (h)	7.3 (6.7–7.8)	7.4 (6.8–7.8)	7.3 (6.7–7.9)	7.3 (6.7–7.8)	7.3 (6.7–7.8)
Number of analyzed epochs (*n*)	737,992	114,522	208,467	177,958	237,045

Values are presented as a number (% of the population) or as a median (interquartile range). BMI: body mass index, OSA: obstructive sleep apnea, AHI: apnea–hypopnea index.

## Data Availability

The dataset analyzed during the current study is not publicly available as it includes medical records and personal information. Thus, the authors are unable to share the data.

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
