# Peer review of "Utilizing Envelope Analysis of a Nasal Pressure Signal for Sleep Apnea Severity Estimation"

_diagnostics, 2023, doi:10.3390/diagnostics13101776_

Round 1

Reviewer 1 Report

Allthough the technique of airflow enevelope analysis was already published in 2015 as a feasibility study in 15 patients (Ciolek et al. 2015) it is up to now widely unknown and underrecognized. With the present study the authors applied the technique to a very large cohort with numerous patients in each category of sleep apnoea (OSA) severity degrees. The aim of the study was to overcome disadvantages of established sleep apnea severity estimation by time and menpower consuming manual scoring of sleep stages and respiratory events (according to AASM rules as international guidelines) without loosing diagnostic quality.

In nasal airflow recordings from 847 suspected OSA patients four parameters were calculated from the difference between the nasal pressure signal’s upper and lower envelopes: average (AV), median (MD), standard deviation (SD), and coefficient of variation (CoV). The results were analysed in their efficiency to distinguish the patients using the conventional apnea-hypopnea index (AHI) thresholds (5-15-30). Additionally, the calculations were done in 30-second epochs to estimate the parameters’ ability to detect manually scored respiratory events. As a result, the SD and CoV were the best classifiers with all AHI thresholds. Furthermore, non-OSA and severe OSA patients were separated well with SD and CoV.

A very important topic is missed to be discussed. The authors did not mention the implications of a differentitation of the respiratory events into central and obstructive types for the choice of the appropriate therapy. They give the information that the algorithm could also be applied to signals of respiratory effort but do not discuss the possibility to differentiate into obstructive and central respiratory events by this further analysis.

The study was done in an appropriate matter. All sections (Introduction, Theory, Materials and methods, Results, and Discussion) are written well and contain all informations necessary in a good informative style of english language. Figures 3 and 6 together with the corresponding text should be deleted because they contain too much too detailed information.

Reviewer 2 Report

I thank the authors and the editor for the opportunity to review this well-written and intriguing paper, which describes the authors' attempts at an alternative method of quantifying OSA severity based on analysis of nasal pressure readings. 

I have no concerns with the authors' study methods nor their analysis, including the statistical methods employed. The results are appropriately presented, and the discussion is proper and appropriate to the findings. The authors' conclusions, that analysis of nasal pressure provides for an alternative means of quantifying sleep apnea severity, appear valid and worthy of further consideration by the field. 

Reviewer 3 Report

The authors used envelope analysis to process nasal pressure signals during sleep as a means to estimate the severity of sleep apnea. This approach is novel and clinically relevant. I have the following comments and questions:

1. In the methods section, it would be great to provide more details about the PSG methods, such as which channels were recorded. The inclusion of thorax and abdomen channels should also be mentioned.

2. The second sentence in section 2.2 was incomplete.

3. How were arousal events (which may cause noise in the signal) dealt with? Were these events excluded during the preprocessing stage?

4. Does the data analysis only include sleep epochs (i.e., excluding wake) or does it include all wake and sleep epochs throughout the whole night?

5. Were bio-calibrations taken into consideration in data analysis? When we manually score the apnea events, technicians may use bio-calibrations as a reference, because an apnea event may appear differently on PSG for different people.

6. The first paragraph on page 4 (starting with "We determined receiver operating characteristic..."): Could the authors please clarify, does the "binary classifications" refer to "has OSA" vs "no OSA"? Or does it refer to the classification of specific OSA severity groups (e.g., "has mild OSA" vs "not mild OSA (which can be no OSA, moderate, and severe OSA)" ? 

7. Figure 4: No respiratory events were identified between epoch ~290 to 430. Was the participant awake during that time? It would be great to include the hypnogram in another panel in the figure. Also, it would be helpful to report the AHI of this participant.

8. Figure 7: Is the y-axis the number of epochs in all participants? Given that the number of participants in each OSA severity group differed, it might be better to plot "the number of epochs per participant" instead as the y-axis.
